# Research Progress of Dendritic Cell Surface Receptors and Targeting

**DOI:** 10.3390/biomedicines11061673

**Published:** 2023-06-09

**Authors:** Chunyu Tong, Yimin Liang, Xianle Han, Zhelin Zhang, Xiaohui Zheng, Sen Wang, Bocui Song

**Affiliations:** College of Life Science and Technology, Heilongjiang Bayi Agricultural University, Daqing 163316, China; tongchunyu@byau.edu.cn (C.T.); liangyimin1314@byau.edu.cn (Y.L.); xianle102123@163.com (X.H.); 13279314802@163.com (Z.Z.); 13359602176@163.com (X.Z.); kcco152@163.com (S.W.)

**Keywords:** dendritic cells, dendritic cell subsets, pattern recognition receptors, targeting, antigen presentation

## Abstract

Dendritic cells are the only antigen-presenting cells capable of activating naive T cells in humans and mammals and are the most effective antigen-presenting cells. With deepening research, it has been found that dendritic cells have many subsets, and the surface receptors of each subset are different. Specific receptors targeting different subsets of DCs will cause different immune responses. At present, DC-targeted research plays an important role in the treatment and prevention of dozens of related diseases in the clinic. This article focuses on the current status of DC surface receptors and targeted applications.

## 1. Dendritic Cells

Antigen-presenting cells (APCs) include DCs, macrophages, B cells, etc. DCs are derived from cord blood CD34^+^ hematopoietic stem cells and bone marrow hematopoietic stem cells. DCs are the most efficient antigen-presenting cells in humans and mammals and are the only antigen-presenting cells capable of activating primary T cells [1]. DCs can cross-present exogenous antigens to MHC class I molecules that help eliminate intracellular pathogens. DCs can express MHC class II molecules quickly and efficiently, and their ability to present peptides is more than 100 times that of other APCs. A single DC can activate 3000 primary T cells. Its role in maintaining immune responses and peripheral tolerance is through antigen capture and presentation of the captured antigen to regulatory T cells [2,3].

### 1.1. Distribution of Dendritic Cells

DCs are widely distributed in various organs and tissues of the human body, except for the brain and eyeballs, and are commonly found in blood, skin mucous membranes, and lymphoid tissues, accounting for less than 1% of the total number of peripheral blood leukocytes [4]. DCs can not only activate a specific immune response but also secrete a variety of cytokines to enhance the body’s immune response [5]. It plays an important pivotal role in primary and secondary immune responses, as well as natural and acquired immunity.

### 1.2. Classification of Dendritic Cells

DCs are differentiated from lymphoid–myeloid hematopoietic stem cells and migrate via body fluids to settle in peripheral tissues, where they differentiate into different subsets. The ability of DCs to accomplish such an arduous task is based on an extensive and complex response mechanism consisting of plasmacytoid dendritic cells (pDCs), myeloid dendritic cells class I (cDC1), and myeloid dendritic cells class II (cDC2). The differentiation and maturation of each DC subset are influenced by the signaling of specific transcription factors, including different levels of IRF8 and IRF4 [6]. Major Histocompatibility Complex (MHC) molecules, which primarily examine, process, and present intracellular and extracellular antigens to T cells, make up the majority of dendritic cells. Surface antigen location and expression were explored to define these two definitions. Traditional/classical myeloid dendritic cell subsets include cDC1, which expresses human CD11c and CD1c/BDCA1, and cDC2, which expresses CD141/BDCA3, Clec9A, and XCR1. Plasmacytoid dendritic cells (pDCs) are class III dendritic cell subsets containing cells expressing CD123, CD304/BDCA4, and CD303/BDCA2 [1,7].

### 1.3. Differentiation of Dendritic Cells

DCs are a very large and complex family of non-lymphoid monocytes with strong heterogeneity and plasticity, and they undergo three stages of differentiation and development, which are generally classified as the precursor, immature, and mature stages. The origin of dendritic cell differentiation is generally divided into two categories: one is the differentiation of myeloid stem cells into dendritic cells stimulated by stimulating factors, and the other originates from lymphoid stem cells and is a heterogeneous antigen-presenting cell (APC) [8]. DCs first differentiate into lymphoid dendritic cells and myeloid dendritic cells during differentiation, and then further differentiate into mDC (Monocyte-DC) and pro-dendritic cells (Pro-DCs). mDC gradually differentiates into peripheral blood dendritic cells, Langerhans cells, mesenchymal dendritic cells, etc., through blood and lymphatic fluid, and their surface markers are CD1a, CD11c, CD13, CD14, and CD33. Pro-DC gradually differentiates into different DC subsets through the blood and lymphatic fluid, and their surface expresses CD123 and BDCA-2.

### 1.4. Maturation of Dendritic Cells

The phenotype, origin, maturation, and function of DCs exhibit a high degree of heterogeneity, with the ability to present DC antigens varying markedly across maturation stages. Blood circulation allows newly differentiated dendritic cells to form into the surrounding tissues, which usually express lower levels of costimulatory and adhesion molecules, and although this does not activate T cells, it promotes the production of T-reg cells, a process important for maintaining homeostasis of the immune state [9].

In general, immature DCs are most abundant in humans and mammals. Although immature DCs have a very weak ability to stimulate mixed lymphocyte responses in vitro, they have a very strong ability to capture and process antigens. Under the influence of various factors such as exogenous antigens and stimulation signals, immature DCs can complete migration from non-lymphoid to secondary lymphoid tissues [10]. Mature DCs have a lower capacity for antigen capturing and processing than immature DCs but can efficiently express adhesion and costimulatory molecules (and integrins) and secrete inflammatory molecules.

It is worth noting that “maturation” and “activation” are different concepts. The term “maturation” refers to the process of DC development from an immature state to a mature state, while “activation” refers to the process of the specific activation of mature DCs after receiving stimulation from external antigens and other substances; that is, the process of transition from immune tolerance to immune response.

## 2. Dendritic Cell Targeting

Since carbohydrates or certain biomolecules used as carriers can be recognized by multiple types of antigen recognition receptors on DCs at the same time, immune specificity and efficacy are significantly enhanced if antigens can be specifically targeted to specific surface molecules on DCs, and the advantages of this strategy are emerging as DC surface molecules continue to be developed, which will play an important role in the production of future vaccines (Table 1).

### 2.1. Significance of Targeting Dendritic Cells

DCs are widely distributed, induce humoral and cellular immunity, and maintain their immune homeostasis by recognizing “foreign” or “self” antigens to generate immune responses and tolerance. A recent study showed that different DC subsets behave differently in regulating the adaptive immune response [23]. Elisa identified CD36 as a new potential target for immunotherapy and showed that one can alter the outcome of the immune response by targeting different receptors on CD8^+^ DCs [24]. Therefore, by using specific surface molecules on different DC subpopulations as targets for specific targeting, we can theoretically manipulate each DC subpopulation directly and thus regulate or control the direction of the immune response by exploiting the specificity of DCs. DCs have powerful immune activating functions and natural adjuvant characteristics, which, with continued research, make DCs useful in improving the efficiency of vaccines, improving the immunogenicity of vaccines, and making the role of DCs in clinical immunotherapy increasingly important.

### 2.2. Targetable Dendritic Cell Subsets and Roles

#### 2.2.1. Plasmacytoid Dendritic Cells

Plasmacytoid dendritic cells (pDCs) have characteristics similar to plasma cells with abundant Golgi apparatuses and endoplasmic reticula for the synthesis of type I interferons, which are the most effective responders to nucleic-acid-based pathogens; thus, in vivo, they play an important role in the surveillance of pathogens from bloodborne sources and autoantigens, among others [25]. When the organism encounters RNA virus or DNA virus infestation, a pDC selectively expresses TLR7 and TLR9, and their signaling pathway depends on multiple factors. When we target ligands such as Ly49Q and CD300a to the ITIM signaling pathway, it enhances the TLR7/9 signaling pathway. When we target RNA-related ligands to FcεR1α, ILT7, BDCA2, NKp44, and Siglec-H receptors to activate them, they negatively regulate the TLR7/9 signaling pathway, while interferon-α-induced pDC-TREM enhances the TLR7/9 signaling pathway [26].

These cells were originally found in human blood and tonsils. Unlike myeloid DCs, which do not express CD11c, CD33, CD11b, and CD13 myeloid antigens, they express GMDP-tagged CD123 (IL-3R) and CD45RA, but their expression is downregulated when DC progenitor cells differentiate into myeloid DCs. Additionally, pDCs with human markers CD303 (CLEC4C; BDCA-2), CD85K (ILT3), CD304 (Neuropilin; BDCA-4), CD85g (ILT7), and characteristic antigens FcR1, BTLA, DR6 (TNFRSF21/CD358), and CD300A can participate in the regulation of physiological functions of the body by producing type I interferon. In his study, Dzionek pointed out that BDCA-2 is a pDC-specific transmembrane lectin with a dual function of trapping Ag and inhibiting IFN-α/β induction, and several recent articles have also elaborated that a small fraction of CD123 pDCs can express CD2, CD56, or CD5 [27,28,29].

Dendritic cell immune receptor (DCIR) is a CLR with immunosuppressive functions, but it lacks dectin-1, mannose receptors, and non-ICAM-3 DC-specific integrins [30,31]. Thus, the antigen is targeted directly to the pDC via CD303 (CLEC4C; BDCA-2) or CD367 (CLEC4A, DCIR) to complete the process. It does not occur through the three mentioned receptors and does not cross-stimulate specific CD8^+^ T cells.

#### 2.2.2. Classical Dendritic Cells

Most DCs in skin, mucosa, and organs, called classical (or conventional) dendritic cells (cDCs), respond to microorganisms by migrating to lymph nodes, where they can process and present protein antigens from microorganisms to T lymphocytes. cDCs can be divided into two classes: class I classical dendritic cells (cDC1) and class II classical dendritic cells (cDC2). Human bone marrow cDC1 exists in blood and tissues in a steady state, and its abundance is about 10 times lower than that of cDC2. It is a subset of circulating DCs and has a high expression of CD141^+^ (BDCA-3, thrombomodulin). Both cDC1 and cDC2 express CD13 and CD33, but cDC1 differs in that it does not have the same low expression as CD11c, CD11b, and SIRPa (CD172) [32,33]. Figure 1 shows the targetable dendritic cell subpopulations and their differentiation processes.

The development of conventional type 1 dendritic cells (cDC1) relies on the presence of BATF3 and IRF8, which facilitate efficient cross-presentation through CLEC9a. These cells also secrete TNF-α, IFN-g, and IFN-1, with a preference for the latter over IL-12p40 upon TLR3 binding [34]. cDC1 cannot rely on CD141 expression alone when identifying cDC1, as the other two cell types also have medium levels of expression in vitro, and in identification, other surface markers should likewise be taken into account, such as CLEC9A, actin receptor during necrosis, CADM1 cell adhesion molecule (NECL2), and BTLA antigen.

Dendritic cells are critical for initiating a protective T cell immune response against viral infection, but viruses can directly infect dendritic cells, disrupting dendritic cell viability and the ability to activate an immune response. cDC1 was reported by Lukowski et al. to be innately resistant to a variety of enveloped viruses, including HIV and influenza viruses, compared to cDC2, which is more susceptible to viral infection. Although this allows viruses to produce an antigen, it impairs their immune function and viability [35]. Resistance to cDC1 infection is conferred by RAB15, a vesicle transporter constitutively expressed in DC subsets. The mechanism of separating viral infection from antigen presentation protects the ability of DCs to initiate adaptive immune function in response to viral infection.

Many common pathways in cDC1 enable effective viral and intracellular antigen detection, antigen transport to the proper endosomal compartments, and type III interferon production to thwart productive viral infection. The CLEC9A receptor recognizes filamentous actin exposed during necrotic cell death, facilitating the cross-presentation of antigens associated with necrotic cells. Zhang et al. have determined the crystal structure of the C-type lectin structural domain of human CLEC9A and proposed a functional dimeric structure with a conserved tryptophan at the ligand recognition site [36]. They suggested that CLEC9A provides targeted adaptive immune system recruitment during infection and could be used to enhance vaccine-generated immune responses. Subsequently, Jessica et al. targeted the antigen to CLEC9A and successfully produced a strong humoral response in mice [37]. Thus, by targeting CLEC9A, the efficiency of vaccines against infectious or malignant diseases can be improved [38,39].

In addition, cDC1 expresses the XCR1 chemokine receptor, which interacts closely with activated T cells and natural killer cells that produce XCL. A CD8+ T cell antigen-driven activation site requires the cross-presentation of XCR1 DCs to initiate primordial CD8+ T cells [40]. Activated CD8^+^ T cells induce XCR1 chemokine-receptor-expressing DCs locally by secreting chemokines XCL1. In their study, the researchers focused on a specific subset of dendritic cells (DCs) known as CD103^+^ CD11b^−^, which express the chemokine receptor XCR1 in the intestine, Tomokazu et al. combined transcriptome and surface marker expression analysis. They hypothesized that T-cell-derived XCLL promotes the activation and migration of small intestinal XCR1^+^ DCs, which in turn provide support for T cell survival and function [41]. CD8+ T cells reorganize local DC networks by interacting with XCR1 DCs, allowing them to better mature and cross-present XCR1 DCs.

Myeloid cDC2 populations isolated from blood, tissues, and lymphoid organs express CD1c, CD2, FcεR1, SIRPA, and the myeloid antigens CD11b, CD11c, CD13, and CD33. According to recent transcriptome analysis, neither CLEC10A (CD301a), VEGFA, nor FCGR2A (CD32A) showed a cDC1 signature consistent with cDC2 [42]. As with pDC and cDC1, myeloid cDC2 develops in response to a variety of transcription factors, but unlike pDC and cDC1, no single transcription factor dominates the process.

cDC2 can be stimulated and thus express IL-12 efficiently while turning into excellent cross-presenting cells. Similar to IL-12p70 secretion, the ability to cross-present exogenous antigens is not limited to cDC1, but cDC2 and pDC can also cross-present [43]. Blood cDC2 has the strongest DC-induced CD8^+^ T cell response in the absence of additional stimuli. cDC2 cross-presents antigens more efficiently than other blood DC subpopulations after in vitro activation [44]. Compared to other antigen-presenting cells, cDC2 is better able to secrete large amounts of IL-12 and can efficiently cross-present antigens and initiate CD8^+^ T cells, inducing a large number of cytotoxic molecules, and in most cases, its ability to synthesize IL-12 is superior to that of cDC1. They secrete IL-23, IL-1, tumor necrosis factor-α (TNF-α), IL-8, and IL-10, but the ability to secrete type III interferons is consistently low. Human cDC2 promotes a broad immune response in vitro, activating Th1, Th2, Th17, and CD8^+^ T cells [45,46].

Bone marrow cDC2s are rich in lectins, toll-like receptors, nod-like receptors, and rig-like receptors, and circulating cDC2s are activated by lipopolysaccharide, flagellin, polyIC, and R848. Among the lectins, high levels of CLEC4A (DCIR/CD367), CLEC10A (CD301), CLEC12A (CD371), and the picric glycoprotein receptor were detected. Similar to monocytes, TLR2, 4, 5, 6, and 8 are present and clearly expressed, as are NOD2, NLRP1, NLRP3, and NAIP [47]. The expression of dectin-1 (CLEC7A) and dectin-2 (CLEC6A) in tissue cDC2 suggests that myeloid cDC2 might be involved in the recognition of fungi.

The CD1 family plays a very important role in resisting the invasion of mycobacterial (belonging to the actinomycetes) antigens into the organism, and CD1a, b, and c involve different molecular mechanisms in capturing different kinds of autoantigens and mycobacterial antigens [48]. CD1a and CD1c elicit a good response when we present glycolipid antigens of mycobacteria and other pathogens through them, but their potential in this regard is often overlooked.

The most studied molecule of CLRs in antibody-mediated dendritic cell targeting is DEC205 (CLEC13B; CD205), which belongs to the mannose receptor family, but the specific ligands for this receptor have not been well studied so far. If the antigen is targeted to DEC205, rabbit anti-mouse DEC205 antibodies are found to be efficiently presented, demonstrating the effectiveness of the dendritic cell targeting strategy in inducing combined pulmonary cellular and humoral immunity, which would be further developed into a highly effective pneumonic plague vaccine for humans [49].

#### 2.2.3. Other Potentially Targeting Dendritic Cell Subsets

Dendritic cells similar to Langerhans cells (LCs) can be found in complex squamous epithelia and the basal epidermis. Like myeloid cDC2, LCs also secrete the C-type lectin langerin and the MHC class I molecule CD1a. Langerin expression is low on myeloid cDC2, whereas LCs express high levels of Langerin, CD1a, and EpCAM, as well as CD11c, CD11b, and CD13, which is a key factor in distinguishing the two classes of cells [50]. In terms of whole dendritic cells, a unique feature of LCs is that they can regenerate independently of bone marrow in the context of primitive and fetal liver hematopoiesis. Langerhans cells form a self-renewal network in mice when PU.1RUNX3 and ID2 bind to locally produced cytokines IL-34 and TFG-β [51]. Two waves of nascent myeloid precursors are recruited during severe inflammation; inflammation leads to a classical population of monocytes followed by a self-renewing network of featureless myeloid precursors [52]. LCs lose their connection to the surrounding epithelium when the skin becomes inflamed, causing them to migrate into afferent lymphatic vessels. In humans, they can mature into effective cross-presenting DCs capable of producing high doses of IL-15 and presenting Mycobacterium glycolipid antigens with the ability to stimulate CD8^+^ T cells [53]. By expressing transgenic human CD1a on mouse LCCs, lipid antigens can be presented to Th17 and Th22 cells. However, Angelic et al. reported that LC lacks a critical TLR, and Julien et al. demonstrated that autologous T cells can be generated from regulatory T cells and IL-22 induced by CD1a-restricted antigens [54,55]. In conclusion, under inflammatory conditions, LCs maintain reactivity to selected intracellular pathogens and viruses, thereby maintaining epidermal health.

Monocyte-derived inflammatory dendritic cells (mo-DCs): As monocyte-derived DCs, mo-DCs exhibit CD13, CD33, CD11b, CD11c, and CD172a expression in inflammation, and according to recent evidence [56], these cells may be recruited from monocytes by their expression of S100A8/9 and CCR2. There is no evidence that monocytes are isolated from DCs based on CD11c and MHC class II expression. Expression of CD1c, CD1a, FcεR1, IRF4, and ZBTB46 could be evidence that monocytes are differentiated from DCs [57]. Inflammatory dendritic cells express CD206 and CD209 while maintaining CD14 expression, and co-expression of CD16, CD163, and FXIIIA has also been reported. However, they showed more characteristics of macrophages than CD1c in the absence of CD1c.

## 3. Pattern Recognition Receptors

Pattern recognition receptors (PRRs) are surface receptors on dendritic cells, including C-type lectin receptors (CLRs), toll-like receptors (TLRs), Fc receptors, and the integrin CD11c. They are often used as target receptors for targeting because of their direct involvement in antigen phagocytosis and presentation and their ability to secrete cytokines that activate the acquired immune response [58].

### 3.1. C-Type Lectin Receptors (CLRs)

C-type lectin receptors belong to the lectin family and represent a class of primary structures with similar carbohydrate recognition regions (CRDs), which are considered to be among the pattern recognition receptors that elicit the body’s immune response to pathogens. A variety of C-type lectin receptors are distributed on various subtypes of dendritic cells, and they are extensively involved in various immunoregulatory processes. C-type lectin receptors are classified into two types. Type I C-type lectin receptors, such as MR/CD206 and DEC205/CD205, target dendritic cells by utilizing their N-terminus located outside the cell. This new method of targeting dendritic cells via the transnasal pathway improves the immune function of mucosal cells to the vaccine. The study’s results suggest that targeting DEC-205/CD205 is a potential new pathway for developing mucosal vaccines against pneumonic plague by activating dendritic cells [49]. Type II C-type lectin receptors express dectin-1, DC-SIGN/CD209, with the N-terminus located intracellularly. HIV-1 infestation and transmission in the colonic mucosa CD209 (DC-SIGN) was found to contribute to the transmission of HIV-1 [59]. Since different subpopulations of dendritic cells have different C-type lectin receptors, different dendritic cell subpopulations have different patterns of antigen uptake, processing, and presentation, so targeting different C-type lectin receptors could theoretically allow antigen binding to different subtypes of dendritic cells, and we could thus regulate the direction and type of immunity. For example, the ability of Y. pestis to release core oligosaccharides and the direct interaction between core oligosaccharides and langerin facilitate the entry of Y. pestis into the LCS, while Y. pestis interacts with Yersinia APCs through langerin. The combination is associated with and may facilitate its transmission and infection [52].

DNGRI (ClecgA) exhibits very strong dendritic cell specificity and is expressed in the CD8^+^ DC subpopulation in mice and BDCA-3DC in humans. Since its discovery in 2008, it has been a hot topic for dendritic cell targeting research because of its restricted expression and phagocytic ability. Using the CD40 antibody as an activator of dendritic cells, Ohe et al. [60] targeted the antigen to DNGRI, which can strongly elicit T cell responses. Antibody yields were increased 100–1000 times compared to untargeted cells against two molecules expressed on the surface of immature mouse DC8 CD8, FIRE (F4/80-like receptor), or CIRE (C-type lectin receptor). In contrast, targeting CD205 is predominantly expressed on CD8+ DCs and does not elicit antibody responses except in an adjuvanted setting.

Langerin/CD207 is expressed in mice mainly in immature Langerhans dendritic cells and is a member of the type II family of CLRs, which are endocytic receptors and are also expressed in humans. Langerin can bind to a variety of pathogens such as Candida albicans which is achieved by recognizing a variety of recognition sites such as mannose, sulfated polysaccharides, etc. Juliana et al. used a new anti-Langerin/CD207 antibody in their study, inserting OVA into the C-terminus of the langerin/CD207 monoclonal antibody, targeting dendritic cells and resulting in the activation of OVA-specific CD4^+^ T and CD8^+^ T cells, and experimentally verifying that Langerin/CD207 can effectively mediate antigen internalization and delivery [61].

The mannose receptor has five structural domains and is a transmembrane protein. Its five structural domains are the amino-terminal cysteine-rich region CR, the type II fibronectin-like repeat region FNII, the transmembrane region, the intracytoplasmic carbohydrate-terminal structural domain, and eight contiguous lectin-like carbohydrate recognition domains CRD. It is through the CRD region that the mannose receptor binds sugar molecules, but it requires the involvement of calcium ions. Each structural domain is capable of binding a mannose, and when these eight structural domains are clustered together, they are more suitable for binding glycans. There are also slight differences among the eight CRDs; for example, CRD4 has a higher binding efficiency when binding D-mannose. Mannose receptors bring antigens to the maturation zone of B cells by binding to splenic lymph node cell surface molecules and presenting antigens to B cell follicles. Mannose-receptor-mediated antigen presentation to dendritic cells has the advantages of selectivity, efficiency, and circularity, and the presentation efficiency can be increased up to 10,000 times if the foreign antigen undergoes presentation by the mannose receptor [47].

### 3.2. Toll-like Receptors (TLRs)

TLRs are capable of recognizing PAMP or its analogs and belong to the PRR family. Since the validation of toll-protein-mediated intrinsic immune responses in Drosophila by Hoffmann in 1996, 12 TLRs have been identified in mice and 10 in humans. Ligands stimulate TLRs to activate the secretion of cytokines such as TNF-α, IL-1, and IL-6 by presenting cells. They demonstrated that by using different TLR agonists as adjuvants, neutrophil sterilization activity can be activated to enhance the resulting memory CTL. This approach leads to a similar protective effect against pathogen attack without any immune-attenuating effects [62].

### 3.3. Fc Receptors

Mouse dendritic cells can express different isoforms to recognize immunoglobulins. The FcγR, FcαR, and FcεR, Fc receptors bind to the long chains of immunoglobulins, and FcγRI, FcγRIII, and FcγRIV are activated in mice. Martin identified a new inhibitory factor called FcγRIIB in his experiments. FcγRIIB is the only inhibitory FcγR identified so far. FcγRIIB has only one chain and the inhibitory signal is transmitted through its intracellular ITIM, which has an important role in the regulation of humoral immune tolerance [63]. Separate targeting of an antigen and antibody to FcγRI results in different effects, activating a specific CTL response if the antigen is targeted to FcγRI and producing a strong humoral immune response if the antibody is targeted to FcγRI. When both activating and inhibiting FcγR are present on the DC surface, there is a neutralizing effect between the two signals, and the summation between the signals will determine whether the antigen activates a cellular or humoral immune response [64].

### 3.4. Integrin CD11c

CD11c belongs to the integrin family, which is one of the major signature molecules of dendritic cells and is expressed at high levels in all subpopulations of dendritic cells. Targeting specific monoclonal antibodies to dendritic cells via CD11c can rapidly elicit antigen-specific humoral immune responses, and CD11c can also induce cellular immune responses, making CD11c an effective immune targeting site [65,66].

## 4. Discussion

Dendritic cells (DCs) are crucial antigen-presenting cells in the immune system responsible for both natural and adaptive immunity. Targeting specific DC subpopulations has been shown to enhance certain types of immune responses and improve disease resistance. Therefore, understanding how to target DCs is essential in developing effective immunotherapies.

For example, the CD11b^+^ DC subpopulation typically exhibits a strong pro-inflammatory response [67], so therapeutic strategies targeting this subpopulation can help enhance body-specific antibodies and cell-mediated immune responses. In contrast, the CD103^+^ DC subpopulation usually plays an important role in the autoimmune response and immune tolerance [68], so therapeutic strategies targeting this subpopulation can help slow down autoimmune diseases and the body’s destructive response to its tissues. Several novel immunotherapies are also being developed to target specific DC subgroups to further stimulate or suppress the immune response. For example, inhibitors targeting hematopoietic factor 2 may inhibit the production of CD8^+^ DC subpopulations, thereby reducing graft rejection and the development of autoimmune disease [69]. There are also DC agonists under investigation, such as antibodies generated through interaction with DC-expressed complex inhibitors (ICOSs), which could enhance the interaction between T cells and DCs and strengthen the cell-mediated immune response [70].

Various DC cell surface receptors have been identified as significant targets for DC vaccines. Clinical trials are currently evaluating receptors that are highly expressed by DC subsets, such as DEC205, CLEC9A, and DEC205, for their potential in cancer immunotherapies, with promising results.

After identifying the target, questions arise about the route of antigen delivery. Conventional DCs (cDCs) are further classified into cDC1 and cDC2, and they possess a strong ability to present antigens. Specifically, cDC1 has the exceptional capability to process endocytosed antigens for MHC-I cross-presentation and activate CD8^+^ T cells efficiently. On the other hand, cDC2 is more proficient in presenting MHC II to CD4^+^ T cells.

Part of the reason for the slow development of dendritic-cell-targeted drugs, despite the tremendous progress made, is that applying this technology to humans is itself a slow process. The advanced nature of the technology dictates that priority be given to human applications, but we can first apply the technology to other fields, such as animal husbandry, which has the advantage of reducing the application cycle of the technology, prioritizing the application of the technology to the clinic, identifying the shortcomings of the technology early or making up for them, and improving it early, which can accelerate the development of the technology. The slow development of vaccines in the livestock breeding industry can be attributed to their high cost, which makes cheaper biochemical reagents more appealing. However, the emergence of dendritic cell vaccines presents an opportunity to reduce the cost of immunization while also improving its effectiveness. Successful development of dendritic cell vaccines can not only improve immunization results but also prevent some pathogens that may arise from the misuse of biochemical reagents. This trend towards dendritic cell development has been observed in recent years.

## Figures and Tables

**Figure 1 biomedicines-11-01673-f001:**
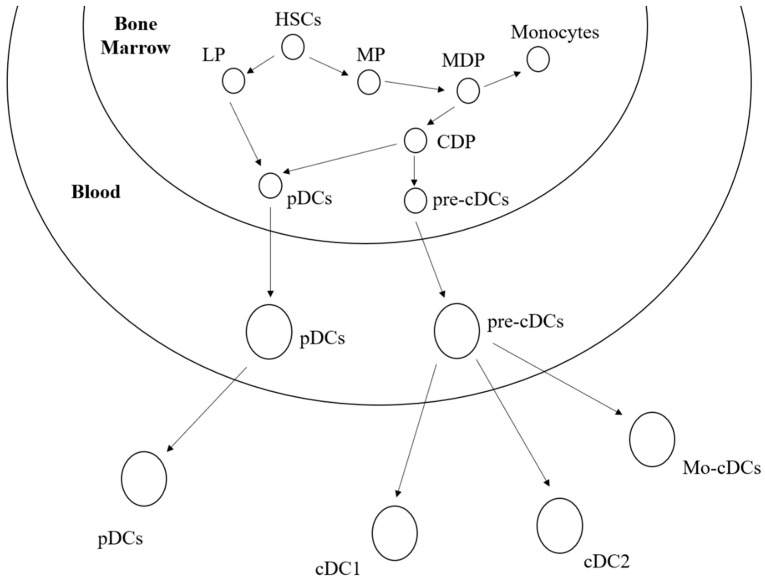
Macrophages (MPs) and lymphoepithelial cells (LPs) are differentiated from hematopoietic stem cells in the bone marrow. Then, MPs successively differentiate into Monocyte-Dendriticcellprogenitors (MDPs), monocytes, and DC progenitor cells (CDPs). Plasmacytoid dendritic cells (pDCs) are derived from LPs and CDPs, and are differentiated from CDPs by classic DC precursor cells (pre-cDCs) followed by the migration of pDCs and pre-cDCs into the blood, where the pre-cDCs differentiate into different types of DC cells (cDC1, cDC2, and Mo-cDCs), and pDCs, cDC1 and cDC2, are distributed in blood, lymphoid, and non-lymphoid tissues, while Mo-cDCs are distributed in inflammatory tissues.

**Table 1 biomedicines-11-01673-t001:** Clinical trials of in vivo DC (antigen-presenting cells (APCs)) targeting cancer with published results.

DC (APC) Targeting Method	Condition	Trial Phase	Trial ID	Refs.
MR targeting	Different advanced cancers	I	NCT00709462; NCT00648102	[11]
DC-SIGN targeting	Different NY-ESO-1-expressing tumors	I	NCT02122861	[12]
DC-SIGN targeting	Melanoma	I	N/A ^1^	[13]
DEC-205 targeting	Different advanced cancers	I	NCT00948961	[14]
SR targeting with HSP	Pancreatic adenocarcinoma	I	N/A	[15]
SR targeting with HSP	Glioblastoma	I	NCT02122822; ChiCTR-ONC-13003309	[16]
SR targeting with HSP	Glioblastoma	II	NCT00905060	[17]
SR targeting with HSP	Glioblastoma	II	NCT00293423	[18]
SR targeting with HSP	Cervical intraepithelial neoplasia III	II	NCT00075569	[19,20]
SR targeting with HSP	Melanoma	II	N/A	[21]
SLOs ^2^ DCs targeting	Melanoma	II	N/A	[22]

^1^ N/A—not available; ^2^ SLOs—secondary lymphoid organs.

## Data Availability

The data are available upon request from the corresponding author.

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
