# Peer review of "Research Progress of Dendritic Cell Surface Receptors and Targeting"

_biomedicines, 2023, doi:10.3390/biomedicines11061673_

Round 1

Reviewer 1 Report

Dendritic cells (DCs) are highly efficient antigen-presenting cells that activate primary T cells to help eliminate intracellular pathogens via both MHC class I and II mechanisms.  This review provides a highly comprehensive and informative summary of multiple aspects of our understanding of DCs with emphasis on their classification, differentiation and maturation.  The review also clearly delineates the cell surface markers that characterize the different classes of DCs. 

This is considered a highly impactful review, which provides an impressively detailed and comprehensive summary of the state of DC research.  The authors wisely use this information to set out a detailed road map to guide the targeting of these specific surface molecules on different DC subpopulations in order to manipulate them and make it possible to regulate the direction of the immune response to specific pathogens.  This could provide tremendous enhancement of the immunogenicity of specific vaccines.  There are no obvious weaknesses in the review. 

The only very minor criticisms refer to the citations of several of the literature references.  As far as I can tell, the names cited do not correspond to any of the authors listed.  The following are just three examples:

1)    Line 254: reference #20 is cited as Drs. Angelic and Julien

2)    Line 310: reference #24 is cited as Juliana

3)    Line 340: reference #27 is cited as Dr. Martin

Please check the citations for all the references, as this may apply to others.

There are only minor problems with the use of the English language.

Author Response

Point 1: Line 254: reference #20 is cited as Drs. Angelic and Julien

Response 1: New references 54 and 55 were re-cited as Angelic and Julien

Point 2: Line 310: reference #24 is cited as Juliana

Response 2: New references 62 were re-cited as Juliana

Point 3: Line 340: reference #27 is cited as Dr. Martin

Response 3: New references 64 were re-cited as Martin

Reviewer 2 Report

Following are my comments for the manuscript:

1) Good beginning of the introduction but there is no reference from lines 18-28 which is entire section.

2)The entire section i.e., line 18-51 is well structured but with only 2 references which is very few.

3) Nicely described classification, differentiation and maturation of DCs but without adequate references. Entire section 1.3 and 1.4 lacks references. Need to sight more literature. 

4) Table 1 is good summary of recent work

5) Figure 1 is nicely described with respect to differentiation of various DC subtypes in different organs but it does not match the statement made in line 100-101 by authors

6) Nice description of section 2.2.1 with targetable DCs but again lack enough references. Line 140 has been written with Prof name which is incorrect style of referencing

7) Line 142 mentions recent studies but without referencing which recent studies

8) Section 2.2.2 lines 149-165 has no references

9) Line 201 mentions recent transcriptome analysis but there is no reference to support which analysis is it

10) Section 2.2.3 lines 238-246 , 248-254 and 259-267 lacks references; Line 261 mentions recent evidence but no references

11) Overall good description of CLRs in section 3.1

12) No references cited for lines 346-349 and 350-355 for sections 3.3 and 3.4

13) First five lines of discussion is repetitive; overall very waivered discussion with no reference whatso ever; Discussion needs to be re-written summarizing efforts in the field and its limitations citing adequate references

1) In line 145, references cited are not superscripted which is inconsistent with previous citations

2) Line 148 mentions Dr. as abbreviation which is incorrect method of referencing

3) In 183, CLEC9A is written in lower case which is an error

4) Language error in lines 300-304 and it needs to be modified as it is referenced to someone's work

5) In line 320 there is a typo by duplication of word "Each"

6) In line 332, there is case error in beginning of sentence "Ligands"

7) Line 341 has language errors and typos

8) Line 368-370 has language errors; message can not be conveyed properly

9) Lines 388-395 has language errors along with typo (missing full stop just before the last statement)

Round 2

Reviewer 2 Report

Most comments made in my previous review has been addressed, however, there is a case error in newly written Discussion for CLEC9A on line 389. It should be in uppercase. 

Author Response

Point 1: Most comments made in my previous review has been addressed, however, there is a case error in newly written Discussion for CLEC9A on line 389. It should be in uppercase. 

Response 1: Clec9a in line 389 was modified to CLEC9A